# Occurrence of *Vibrio* spp. and *Pseudomonas* spp. Isolates of *Nodipecten nodosus* (Linnaeus, 1758) and Water from a Mariculture Farm in Angra dos Reis, Brazil

**DOI:** 10.3390/microorganisms13040752

**Published:** 2025-03-26

**Authors:** Antônia L. dos Santos, Salvatore G. De-Simone, Guilherme S. L. Carvalho, Kayo C. B. Fernandes, Maysa B. M. Clementino

**Affiliations:** 1Center for Technological Development in Health (CDTS)/National Institute of Science and Technology for Innovation in Neglected Population Diseases (INCT-IDPN), Oswaldo Cruz Foundation (FIOCRUZ), Rio de Janeiro 21040-900, RJ, Brazil; antonia.lucia@fiocruz.br; 2Epidemiology and Molecular Systematics Laboratory (LEMS), Oswaldo Cruz Institute, Oswaldo Cruz Foundation, Rio de Janeiro 21040-900, RJ, Brazil; 3Program of Post-Graduation on Science and Biotechnology, Biology Institute, Federal Fluminense University, Niteroi 24020-141, RJ, Brazil; 4Program of Post-Graduation on Parasitic Biology, Oswaldo Cruz Institute, Oswaldo Cruz Foundation, Rio de Janeiro 21040-900, RJ, Brazil; 5National Institute of Infectious Diseases Evandro Chagas (INI), Oswaldo Crus Foundation (FIOCRUZ), Platform for Identification of Microorganisms by Mass Spectrometry, Rio de Janeiro 21040-900, RJ, Brazil; guilherme.silva@ini.fiocruz.br; 6Reference Microorganisms Laboratory, National Institute for Quality Control in Health (INCQS), Department of Microbiology, Oswaldo Cruz Foundation (FIOCRUZ), Rio de Janeiro 21045-900, RJ, Brazil; kayo.bianco@fiocruz.br (K.C.B.F.); maysa.mandetta@fiocruz.br (M.B.M.C.)

**Keywords:** marine pollution, marine bivalves, antimicrobial susceptibility, *Pseudomonas* spp., *Vibrio* spp.

## Abstract

Bivalve mollusks face a crisis due to infectious diseases, resulting in high mortality and economic losses. The need for continuous monitoring to prevent contamination from sewage and rainwater in aquaculture is evident. The recent mass mortality of scallops in Ilha Grande Bay (IGB), Rio de Janeiro’s largest scallop producer, due to environmental contaminants underscores the need for further research. This study aims to investigate the recent collapse of the scallop population and assess the human impact by analyzing the circulation of pathogens. Materials and Methods: Mollusks were collected from three sites in Ilha Grande Bay (IGB), a region known for its significant scallop production, and from scallop farms in Angra dos Reis, RJ. A total of 216 gill and adductor tissue samples from lion’s foot scallops were analyzed. Bacterial contamination was identified using MALDI-TOF, while antimicrobial susceptibility and carbapenem production were assessed via disk diffusion tests. Results: Mollusks were contaminated with *V. alginolyticus*, *V. fluvialis*, *V. harveyi*, *Pseudomonas putida*, and *Pseudomonas monteilii.* All isolates were sensitive to meropenem, but *P. putida* showed higher resistance to ciprofloxacin. Conclusions: The presence of these pathogenic and resistant bacteria in scallop adductor tissues is a concern for the aquaculture industry and a significant public health risk. The potential for these bacteria to enter the human food chain through consuming contaminated seafood or recreational activities such as bathing is a serious issue that needs to be addressed.

## 1. Introduction

Given the increasing need for food production to meet global needs, aquaculture provides a sustainable alternative to meet the demand for seafood, offering high nutritional value and conserving natural resources. Therefore, this relevant socio-economic activity is expanding and increasing globally for the cultivation, production, and marketing of food as an alternative to reduce pressure on overfished populations [1]. On the other hand, pollution in general, especially by sewage, threatens the health of seafood, increasing its vulnerability to diseases and reducing productivity [2].

The scallop *Nodipecten nodosus* (Pectinidae) is a globally significant bivalve in mariculture in several parts of the world, including Brazil, and it has high culinary value worldwide.

In 2022, production reached 1.74 million tons, primarily in China, the USA, and the EU, generating USD 4.3 billion in revenue (2.8% of aquatic products) [3,4]. Growing environmental concerns in commercial production highlight the need to ensure scallop health and growth. These bivalves are economically relevant in aquaculture and fisheries. Still, they are susceptible to environmental changes such as the acidification of the ocean [5] and, as benthic filter-feeders, aquatic invertebrates are vulnerable to contaminants such as heavy metals [6], viruses [7], bacteria, and protistans [8,9,10], which are responsible for mortality outbreaks and have a substantial commercial impact [11,12]. To a lesser extent, other diseases are also caused by fungi (*Aspergillus*, *Penicillium*, and *Fusarium*) [13,14,15], Porifera (*Cliona* spp.) [16], and helminth parasites, such as trematodes, cestodes, and nematodes [17,18], and, therefore, determinants of transmission represent a public health challenge.

On the other hand, microbial diversity and contamination levels can vary due to organic matter and lower oxygenation, which favors the proliferation of anaerobic microbes [19,20]. Another critical point is the current and climatic events that irregularly disperse sewage and industrial waste, further exacerbating bacterial growth and increasing bivalves’ susceptibility to contamination [21].

Brazilian mariculture, particularly in Rio de Janeiro, faces challenges due to limited research on biotic and abiotic parameters. The vulnerability of cultured scallops stems from inadequate knowledge of environmental and biological impacts, including bacterial proliferation affecting stock stability and causing economic losses. In Ilha Grande Bay (IGB), Angra dos Reis (RJ, Brazil), rapid population growth (73.9% from 2010 to 2021) has increased pressure on water resources, leading to higher contamination from domestic and industrial effluents. Sampling in affected areas revealed links between scallop mortality and environmental degradation [22].

The microbiological monitoring of aquaculture waters and shellfish in Brazil and other countries primarily relies on bacterial indicators such as thermotolerant coliforms [23], *E. coli*, and *Enterococcus* spp. [24].

Brazil’s third largest coastline, Rio de Janeiro, has suitable ecosystems for bivalve farming, mainly cultivating *N. nodosus*, *Perna perna* (mussels), and *Crassostrea* spp. (oysters). Niteroi and Ilha Grande Bay (IGB) are key production areas; however, in 2022, a decline in shellfish production was reported, which significantly affected the economy of Rio de Janeiro. From 51.2 tons in 2016, the productive mariculture of *N. nodosus* decreased to 10.2 tons in 2022, leading to substantial economic losses [25,26].

Given the continued decline in *N. nodosus* production and increasing mortality rates in Ilha Grande Bay since 2018, reaching approximately 100% in 2021, this species was analyzed in this study. Understanding the factors that influence the growth and mortality of these bivalves is crucial to improving management practices and minimizing losses in aquaculture. This understanding is a key part of the research process and engages the audience in the study. Another critical step is the possibility of multidrug-resistant (MDR) bacteria that pose serious public health problems. The development of MDR is driven by natural genetic variation and bacterial interactions but is accelerated by the indiscriminate use of antimicrobials in medicine and food production.

This study aimed to isolate and identify microbial contaminants in water and *N. nodosus* samples from mariculture areas of Rio de Janeiro (Bananal Island Cove, Gipóia Isle Cove, and Nautica Beach Cove in Jacuecanga). The goal was to evaluate the sanitary conditions of bivalve farming in Ilha Grande Bay, assess their impact on scallop health and quality, and examine their implications for environmental sustainability and public health risks, as defined by existing regulations.

## 2. Materials and Methods

### 2.1. Collection Study Area

Between February and December 2021, 11 collections were carried out on mariculture farms on the southern coast of the State of Rio de Janeiro (Brazil), totaling a collection of 275 scallops (55 L of water) at a depth of 15 to 20 m. Figure 1 shows the geospatial map of the three collection points: Jacuecanga Beach [Figure 1—(1); IED-BIG; 23°00′29″ S and 44°1413″ W], Gipóia Island (Figure 1—(2); 23°02′16″ S and 44°23′08″ W), and Bananal Island (Figure 1—(3); 23006′04 S and 44°15′07″ W). The collections were carried out in partnership with the Instituto de Eco Desenvolvimento da Baía da Ilha Grande (IED-BIG) in the Atlantic Forest, an area of high biodiversity recognized as a World Heritage Site by UNESCO [27].

The criteria for selecting the points for the three collections of scallops *N. nodosus* (the largest native pectinid in Brazil) were based on the mortality impacts observed in recent years. No mortality was described on Gipóia Island, but high larval mortality was detected on Bananal Island and in Jacuecanga Bay.

Mariculture is a prominent economic activity in the region, with cultivation in lantern nets (IED-IGB, 2022; https://iedbig.org.br/), accessed on 12 September 2022). The scallops *N. nodosus* were collected manually and randomly, packaged in glass bottles in groups (*n* = 5) in local water samples (1 L). They were packaged in sterilized polyethylene bags, identified, and stored in thermal boxes (6–10 °C) to ensure integrity (RDC n° 724/2022; https://www.cidasc.sc.gov.br/inspecao/files/2023/11/Resolução-RDC-No-724-DE-1/07/2022, accessed on 12 November 2022) and transported to the laboratory (Microorganism Reference Laboratory of the National Institute of Quality Control in Health (INCQS) and to the Center for Technological Development in Health (CDTS) of the Oswaldo Cruz Foundation) by ISO 194558:2006 standards [28].

### 2.2. Sample Processing and Bacterial Isolation

The samples underwent an initial external wash to remove surface marine materials according to the Interministerial Normative MPA/MAPA criteria for human consumption from 7 May or 8 May 2012 in Brazil [29,30]. They were then immersed in a chlorinated water solution (5 ppm Free Residual Chlorine—CRF), as required by health inspection legislation [31]. Adult scallops (*n* = 275; the size of 6.25 ± 1 cm) had their valves washed externally (to remove residues and under aseptic conditions), opened with a sterile spatula to access the soft tissues, and adductor muscles disconnected.

The branchial arch tissue and adductor muscle were removed with forceps and scalpel. Then, 25 g of each sample was added to 225 mL of buffered saline and homogenized separately in a Stomacher (60 seg; Laborglass, São Paulo, SP, Brazil) for 60 s as described previously [15].

The macerated scallop tissues and collected seawater were filtered on cellulose acetate membranes (0.22 μm; Sigma Chemical Co., Ltd., St. Louis, MO, USA), and all the materials (seawater, scallop branchial arch, adductor muscle tissues, and filtered membranes) inoculated in BHI containing 1–3% NaCl. Then, the samples were cultivated on plates with a solid BHI medium to isolate microorganisms. The isolated strains were grown in BHI agar at 35 °C for 24 h, and representative colonies of each tissue were transferred to a microplate (96 MSP, Bruker, Billerica, MA, USA) using a loop. After drying, 1 μL of 70% formic acid (Sigma-Aldrich, St. Louis, MO, USA) was added to the bacterial sediment to optimize cell lysis.

After drying the lysis solution, 1 μL of matrix solution (α-cyano-4-hydroxycinnamic acid diluted in 50% acetonitrile and 2.5% trifluoroacetic acid (Sigma-Merck, St. Louis, MO, USA) was added.

### 2.3. MALDI-TOF Analysis

Isolated samples were analyzed by MALDI-TOF LT Microflex mass spectrometer (Bruker Optic GMBH, Ettlingen, Germany) equipped with a 337 nm nitrogen laser in linear mode. The operation was controlled by the FlexControl 3.3 software (Bruker Optic GMBH, Ettlingen, Germany). The spectra were collected in the mass range between 2000 and 20,000 *m*/*z* and subsequently analyzed by the MALDI Biotyper 2.0 software (Bruker Optic GMBH, Ettlinger, Germany) using standardized settings for bacterial identification. These studies were conducted in the Bacteriology and Bioassays Laboratory of the Evandro Chagas, National Institute of Infectiology (INI/FIOCRUZ).

### 2.4. Criteria for Classifying Samples

Antimicrobial susceptibility was determined by the disk diffusion technique (Kirby-Bauer method) according to the criteria established by the European Committee on Antimicrobial Susceptibility Testing [32]. The isolates were initially plated on BHI agar for 24 h at 37 °C. The microbial growth was suspended in a sterile saline solution (0.85% NaCl) to obtain a turbidity standard of 0.5 on the McFarland scale. The suspension was plated on Mueller–Hinton agar media with a swab, where the antimicrobial disks were deposited. The inhibition zones were read after incubation at 37 °C for 16–18 h.

Isolates that showed resistance to up to two antimicrobials of the same class were classified as non-MDR (non-multidrug-resistant); resistance to up to three antimicrobials of different courses was classified as MDR (multidrug-resistant); resistance to at least one antimicrobial in all classes—except ≤2—was classified as XDR (extensively drug-resistant); and resistance to all antimicrobials tested was classified as PDR (pan-drug-resistant).

## 3. Results

The MALDI-TOF MS analysis (275 scallops and 55 L of water) detected the presence of 216 species of microorganisms (100%) in scallop tissues. The most prevalent genus was *Pseudomonas* spp. (*n* = 62; 29%), followed by *Vibrio* spp. (*n* = 52; 24%), but other minor genera were also identified, as shown in Figure 2.

Hence, as these two species were the most prevalent, the susceptibility to antimicrobials was analyzed. From 114 isolates selected, 62 (54.4%) were *Pseudomonas* spp. [(52 (45.6%) *P. putida* and 10 (8.8%) *P. monteilii*)] and 52 (45.6%) were *Vibrio* spp. [(37 (32.5%) *V. alginolyticus*, 11 (9.6%) *V. fluvialis*, and 4 (3.5%) *V. harveyi*)].

The results are shown in Figure 3 (*P. putida* and *P. monilia*) and Figure 4 (*V. alginolyticus*, *V. fluvialis*, and *V. harveyi*).

*P. putida* showed a higher percentage of resistance to ciprofloxacin (42%, 22/52), followed by cefepime (37%, 19/52), aztreonam (35%, 18/52), ceftazidime (6%, 3/52), amikacin (4%, 2/52), and piperacillin/tazobactam and imipenem (2%, 1/52). Most P. putida isolates were sensitive to meropenem. Meanwhile, *P. monteilii* isolates showed resistance to aztreonam alone (40%, 4/10), followed by cefepime and ciprofloxacin (10%, 1/10), being sensitive to other antimicrobials evaluated (Figure 3).

The isolates of *V. alginolyticus* and *V. fluvialis* showed high percentages of resistance to tetracycline (70% and 82%, respectively), followed by piperacillin/tazobactam, where *V. harveyi* is also found (54%, 91%, and 25%). Resistance to ciprofloxacin was 41%, 73%, and 75%, and to ceftazidime, 32%, 36%, and 0%. *V. alginolyticus* isolates showed the highest percentage of resistance to tetracycline (70%, 26/37), followed by piperacillin/tazobactam (54%, 20/37), ciprofloxacin (41%, 15/37), azithromycin and ceftazidime (32%, 12/37), meropenem (24%, 9/37), and sulfamethoxazole/trimethoprim (11%, 4/37). *V. fluvialis* isolates showed predominant resistance to piperacillin/tazobactam (91%, 10/11), followed by tetracycline (82%, 9/11), ciprofloxacin (73%, 8/11), meropenem (64%, 7/11), ceftazidime (36%, 4/11), azithromycin, and sulfamethoxazole/trimethoprim (27%, 3/11). In turn, *V. harveyi* isolates showed resistance only to ciprofloxacin (75%, 3/4), piperacillin/tazobactam, and azithromycin (25%, 1/4) (Figure 4).

Considering the medical importance of this study’s findings, isolates of the three species of the genus *Vibrio* spp. and the two species of *Pseudomonas* spp. were classified according to antibiotic susceptibility as MDR, PDR, XDR, and non-MDR according to the recommendations of CLSI and EUCAST. The results are presented in Figure 5.

The *P. monteilii* and *P. putida* isolates were classified as MDR, demonstrating resistance to at least one agent in three antimicrobial categories. Those identified as XDR exhibited resistance to at least one agent in six categories. Isolates that did not show susceptibility to all commercially available antimicrobial agents were classified as PDR [33].

The *P. putida* (84%) and *P. monilia* (16%) showed resistance to antibiotics, with *P. putida* and *P. monteilii* showing the following percentages, respectively: ciprofloxacin (42%, 10%), cefepime (37%, 10%), aztreonam (35%, 40%), ceftazidime (6%, 0%), amikacin (4%, 0%), piperacillin/tazobactam (2%, 0%), meropenem (2%, 0%), and imipenem (2%, 0%).

## 4. Discussion

This study found that *N. nodosus* mollusks from three farms—IED-BIG, Bananal, and Gi Point—were contaminated with antimicrobial-resistant bacteria in the Ilha Grande Basin, Rio de Janeiro, Brazil, and surrounding waters. This is concerning as the region is Brazil’s top scallop producer. The most prevalent species were *Pseudomonas* spp. and *Vibrio* spp.

Mariculture is vital to the BIG economy, but high mollusk mortality—likely due to worsening water quality and climate change—threatens the industry’s stability [17].

The presence of *Vibrio* spp. and *Pseudomonas* spp. underscores human impact on the production area, posing environmental and health risks. Continuous monitoring is crucial. Between 2010 and 2021, the local population grew by 73.9%, yet only 33% had sewage treatment, leading to untreated effluents in the BIG. Limited water circulation worsens pollution, affecting scallop production [9]. Mass mortality events linked to *V. alginolyticus*, *V. fluvialis*, *V. harveyi*, *P. monteilii*, and *P. putida* have already been reported in 2018 in Brazil [34].

In our study, the seawater was filtered through gravel at ~4 m depth, passing through a nozzle, sand filters (10 μ, 3 μ, 2 μ, 1 μ), and UV treatment before reaching the laboratory. Despite these measures, bacteria were detected in IED-BIG’s water and scallops, highlighting the need for stronger biosecurity. Antimicrobial-resistant *P. monteilii*, *P. putida*, *V. alginolyticus*, *V. fluvialis*, and *V. harveyi* were found at all sites [33]. *V. alginolyticus*, the second most common *Vibrio* species globally, is a major aquaculture threat [35].

Benthic pectinids indicate environmental degradation due to climate change factors—ocean warming, acidification, and hypoxia—impacting scallop survival and reproduction [36,37,38]. Rising temperatures, lower dissolved oxygen, and reduced pH have affected species like *Pecten maximus*, *Argopecten irradians*, and *N. nodosus* [39]. Climate change also reduces phytoplankton productivity and increases bacterial infections in scallops [40,41].

Effective monitoring is essential to ensure food security in mariculture. AMR is a growing global concern, causing ~700,000 deaths annually, with projections exceeding 10 million by 2050 [42]. *P. aeruginosa* is among the most critical AMR pathogens [43], with infections being difficult to treat due to low membrane permeability, β-lactamase production, and efflux pump systems. *Pseudomonas* spp. can acquire nearly all known AMR mechanisms, posing a global challenge.

*N. nodosus*, *V. alginolyticus*, and *V. fluvialis* have been observed with high resistance rates to tetracycline (70–82%), piperacillin/tazobactam (25–91%), ciprofloxacin (41–75%), and ceftazidime (0–36%). These findings stress the urgent need for improved biosecurity and wastewater management to mitigate AMR risks.

Our findings align with studies in Iran, Sudan, Egypt, and South Africa [44], which reported *P. aeruginosa* resistance to 14 antimicrobials across eight categories. High resistance rates were noted for ticarcillin/clavulanic acid (89.3%), meropenem (50.9%), and fosfomycin (37.7%). Resistance to antipseudomonal ranged from 37.7% to 14.5%, with the highest rates for cefepime (37.7%), imipenem (37.4%), and aztreonam (25.3%). However, polymyxins and aminoglycosides showed greater efficacy, with susceptibility rates of 90.3% colistin, 86.5% amikacin, and 82.4% tobramycin.

Community-acquired MDR *P. aeruginosa* infections remain rare [45]. A study of 60 patients with community-acquired bloodstream infections found 100% susceptibility to meropenem and 95% to piperacillin/tazobactam and ceftazidime. A Turkish case described an abscess caused by a *P. aeruginosa* strain susceptible only to imipenem, amikacin, and colistin [46].

Preventing MDR bacterial spread is a pressing public health concern, yet regional susceptibility data remain scarce. Even when available, identifying community-linked isolates is challenging. While clinical cases attract policymakers’ attention, subclinical transmission is already occurring [47].

Previous studies [44] have examined *Vibrio* spp. distribution in aquatic environments to assess human exposure risks. Over two years, ten pathogenic *Vibrio* species, including *V. cholerae* O1, were detected in Georgian waters. *Vibrio* spp. is linked to cholera, responsible for seven pandemics and 4 million cases annually, causing up to 143,000 deaths per year [48,49].

*Vibrio* spp. constitute a known foodborne pathogen that poses risks in natural aquatic environments. However, the species of Vibrio responsible for disease in aquaculture settings and their associated virulence genes are often variable or undefined.

Canellas [50] detected the presence of Vibrio spp. at three sites in Guanabara Bay (Brazil), while Hackbusch et al. [51] monitored *V. parahaemolyticus*, *V. vulnificus*, and *V. cholerae* over 14 months in the North Sea, revealing seasonal trends in virulence genes. These findings stress the need for long-term studies to understand bacterial behavior and health risks, as the public health impact of this genus remains poorly understood [52].

The incidence and severity of *V. alginolyticus* infections have risen significantly over the past decade. As an opportunistic pathogen, it has caused major economic losses in mariculture and contributed to the increase in vibriosis cases [53]. While *V. alginolyticus* is the second most common *Vibrio* species globally, other species have also grown in prevalence [54]. In southern China, it ranks third among *Vibrio* species isolated from diseased marine fish, following *V. harveyi* and *V. vulnificus* [55].

Resistance to third-generation cephalosporins in *Vibrio* spp. is frequently documented, particularly in seafood isolates. Lee et al. [56] found *V. parahaemolyticus* in Malaysian fish markets, with 52% and 28% resistance to cefotaxime and ceftazidime, respectively, and 2.4% carrying the *trh* gene linked to pathogenicity. Similarly, a South Korean study reported 5.7% and 8.6% resistance in *V. parahaemolyticus* from fish farms, with over 90% carrying pathogenicity-related genes [57]. Researchers suggest widespread resistance in Asia due to excessive antimicrobial use in aquaculture [58,59].

In China, Yu et al. [60] reported that *V. alginolyticus* triggers epidemic outbreaks in marine species, including fish, shrimp, and mollusks, threatening aquaculture. These outbreaks are most common at water temperatures of 25–35 °C. Infections have been documented in Europe, Asia, and the Americas [60,61,62], often linked to raw seafood consumption or exposure to contaminated seawater [63,64,65].

These findings underscore the urgent need for improved sanitation and wastewater management to prevent the spread of multidrug-resistant, extensively drug-resistant, and pan-drug-resistant bacteria. In Angra dos Reis, *P. putida*, *P. monteilii*, *V. alginolyticus*, *V. fluvialis*, and *V. harveyi* exhibited resistance to cefepime, ceftazidime, and ciprofloxacin. MDR resistance was found in *P. putida*, *V. alginolyticus*, and *V. fluvialis*, while XDR was detected in *V. fluvialis* and *V. alginolyticus*. PDR resistance appeared exclusively in *V. alginolyticus.*

## 5. Conclusions

The study highlights the significant impact of microbial contamination on mariculture in Ilha Grande Bay (IGB), Brazil, particularly due to *Pseudomonas* spp. and *Vibrio* spp. The detection of these bacteria, including antimicrobial-resistant strains, suggests anthropogenic pollution, which is likely worsened by inadequate sewage treatment and poor water circulation. The collapse of scallop production, from 51.2 tons in 2016 to 10.2 tons in 2022, may be linked to environmental stressors such as climate change, water quality deterioration, and bacterial infections. Notably, *V. alginolyticus*, *V. fluvialis*, and *P. putida* demonstrated concerning resistance profiles, including multidrug-resistant, extensively drug-resistant, and pan-drug-resistant classifications. These findings emphasize the urgent need for improved wastewater management, continuous environmental monitoring, and stricter antimicrobial use regulations to mitigate public health risks and support sustainable mariculture.

## Figures and Tables

**Figure 1 microorganisms-13-00752-f001:**
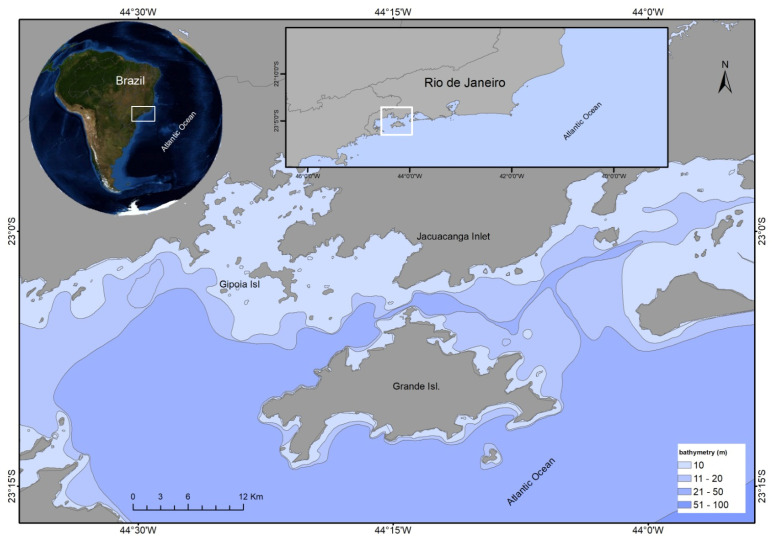
Geospatial map of the three collection points of samples in Rio de Janeiro, Brazil. (1) (Jacuecanga beach, IED-BIG; 23°00′29″ S e 44°1413″ W), (2) (Gipóia isle, 23°02′16″S e 44°23′08″ W), and (3) (Bananal isle, 23006′04 S e 44°15′07″ W). The image was produced using the geospatial platform ArcGIS (program ArcGIS 10.8; https://www.img.com.br/pt-br/arcgis/plataforma-geoespacial/visao-geral#contato&keyword=arcgispercentage20lcenpercentageC3percentageA7a &ad=458363843178%22 (accessed on 22 May 2024)).

**Figure 2 microorganisms-13-00752-f002:**
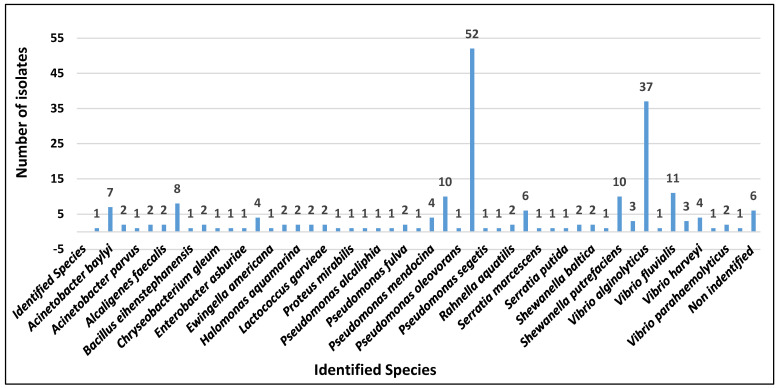
Number of microorganism species identified by MALDI-TOF MS and isolated from tissues of 275 scallops.

**Figure 3 microorganisms-13-00752-f003:**
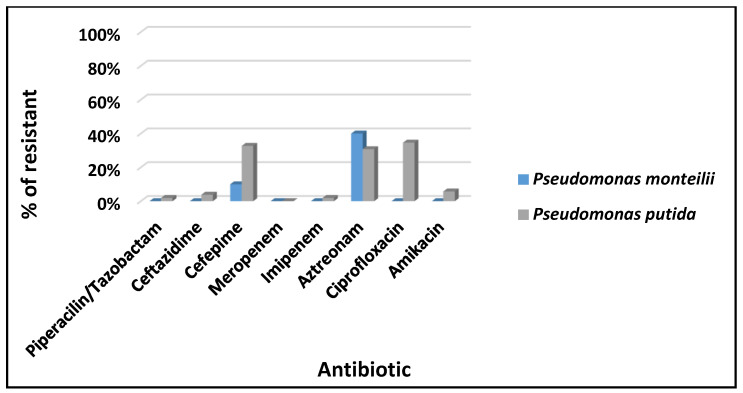
Antimicrobial susceptibility profile of *P. putida* and *P. monteilii* isolates recovered from scallops and water.

**Figure 4 microorganisms-13-00752-f004:**
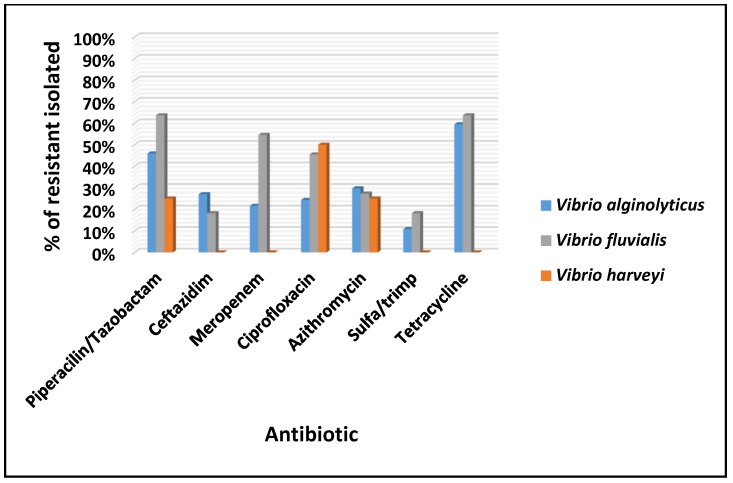
Antimicrobial susceptibility profile of *V. alginolyticus* isolates, *V. fluvialis*, and *V. harveyi* recovered from scallops and water.

**Figure 5 microorganisms-13-00752-f005:**
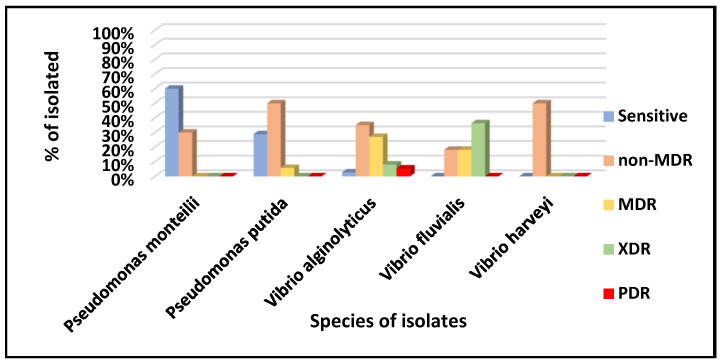
Classification of antimicrobial susceptibility profiles concerning evaluated species.

## Data Availability

The data presented in this study are available upon request from the corresponding author.

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
