# Peer review of "Occurrence of Vibrio spp. and Pseudomonas spp. Isolates of Nodipecten nodosus (Linnaeus, 1758) and Water from a Mariculture Farm in Angra dos Reis, Brazil"

_microorganisms, 2025, doi:10.3390/microorganisms13040752_

Round 1
Reviewer 1 Report
Comments and Suggestions for Authors
Dear Authors,
I appreciate your efforts to research this topic.
Please find below some observations and suggestions.
All the best
- Revise the abstract and instead of just writing the word “novelty” please make a strong statement that represents the novelty of the work.
- Make a major revision of the introduction part. The scientific names especially bacterial names should be italicized.
- Once use abbreviations don’t repeat the full form. For example, in Line 67 and Line 75 the word E.coli.
- The chosen sampling location mentioned in the manuscript why was selected? do they represent the overall contamination risk in the region?
- Samples were gathered at depths ranging from 15 to 20 meters, the report notes. How was this depth range calculated, and may microbial diversity and contamination levels alter dramatically across depths?
- Did the study incorporate any secondary confirmation methods (for example, 16S rRNA sequencing) to ensure the correctness of MALDI-TOF bacterial identification?
- The study classifies isolates as MDR, XDR, or PDR based on resistance patterns. Were any genetic investigations (e.g., PCR, whole genome sequencing) conducted to validate resistance genes and their potential for horizontal transfer?
- Make a general revision to the conclusion parts with significant findings.
- The article needs English proofreading.
The article needs English proofreading
Author Response
Thank you, I appreciate your efforts in researching this topic.
1) Please review the abstract and make a strong statement about the work's novelty instead of just writing the word "novel."
R: This was done.
2) Please make a major revision of the introduction. Scientific names, especially bacterial names, should be in italics.
R: Thank you. The requested corrections have been made.
3) Once abbreviations are used, do not repeat the full form. For example, in Line 67 and Line 75, the word E. coli.
R: Thank you, the requested corrections have been inserted.
4) Did the chosen sampling site mention why it was selected in the manuscript? Do they represent the overall risk of contamination in the region?
R: The criteria for selecting points for the three specific collections were based on the different impacts observed on scallop mortality: Gipóia Island, where there was no mortality; Banana Cove, a mass mortality site; and scallop cultivation, where high larval mortality was recorded.
5) The report notes that the samples were collected at depths ranging from 15 to 20 meters. How was this depth range calculated, and can microbial diversity and contamination levels change dramatically between depths?
R: Seawater is collected through a gravel filter at a depth of ~4m, passing through a nozzle with a valve and a 15mm diameter collector. A 3 HP pump takes it to the laboratory, passing through sand filters, 10 microns, and cartridges (3.2 and 1 micron). It ends with UV treatment, where the matrices are located, indicating the importance of these measures for a more excellent seed production. Studies indicate anthropogenic contamination by antimicrobial-resistant bacteria in aquatic environments in Rio de Janeiro. At IED-BIG, bacteria were detected in the collection water and scallops, even after filtration and UV treatment.
6) Did the study incorporate any secondary confirmation method (e.g., 16S rRNA sequencing) to ensure the accuracy of MALDI-TOF bacterial identification?
R: Antimicrobial resistance is a global challenge that compromises several treatments against bacteria and other pathogens. Rapidly identifying these microorganisms is essential to optimize patient treatment. It can also be applied in agriculture, contributing to safety, ecology, and quality testing (WEIS, 2020). Conventional phenotyping and sensitivity testing methods take around 48 hours, which can aggravate emergency conditions. The MALDI-TOF system enables the rapid and reliable identification of bacteria, yeasts, neoplasms, and fungi and is helpful in clinical diagnoses, environmental research, taxonomy, and food quality control (MAIER et al., 2006; RODRIGO et al., 2014). Although other analyses can complement the data, MALDI-TOF provides sufficient information for an accurate assessment.
7) The study classifies the isolates as MDR, XDR, or PDR based on resistance patterns. Was any genetic investigation conducted (e.g., PCR, whole genome sequencing) to validate resistance genes and their potential for horizontal transfer?
R: Unfortunately, due to several operational difficulties, we were unable to perform the genetic analysis during this period. We found it important, but it was pending for the next work. The results indicate human contamination in the scallop-producing region. In all analyzed points, the presence of Vibrio alginolyticus and Pseudomonas monteii, both highly resistant to antimicrobials and pathogenic potential, represents a threat to aquatic animals and human health. Aquatic monitoring of the analyzed regions is relevant to estimating possible risks to public health. The presence of potential scallop pathogens in seawater and muscles could increase with the warming and pollution of seawater. The study of the genus Vibrio spp. and Pseudomonas spp. offers great potential and should be beneficial, especially when it comes to bacteria isolated from an ecosystem of great ecological, economic, social, and clinical value in BIG, Angra dos Reis, Rio de Janeiro, Brazil
8) The article needs to be reviewed in English.
A: The manuscript was reviewed by a native researcher.

Reviewer 2 Report
Comments and Suggestions for Authors
Dear Authors,
As a reviewer I need to point out, that the current state of the manuscript does not allow for the publication in the scientific journal. I have noticed a number of issues, that need to be dealt with before. At this point it is very visible that the proposed manuscript lacks the proper polish, and quality.
Here I will provide some of the remarks for the specific chapters.
1.Introduction
Overall remarks:
Unnecessary repetitions, some of the sections of the Introduction are repeating the same information. See lines 63-64 and 55-56 as an example.
In some cases the text is lacking seamless linkage between the paragraphs.
Line 45-46 Section on biology of the organism should be expaneded to better highlight the reasons why its biology is making it a potential reservoir for potential pathogens, and alongside them ARG's.
2.Methods section
Overall remark - Different font sizes
Unnecessary bolding of certain subchapters.
Some descriptions should be expanded, as they are just confusing - see line 143 as an example
Names of the chapters are being repeated.
Line 122-á
Line 126 - needs to be corrected
Line 129 - na áre
Line 165 and 167 - no need to repeat about the legislation twice
I dont see anything particulary troubling regarding the MALDI-TOF part of the studies.
3.Results
Once again a section of the text has been repeated, information - "This study evaluated the microbial quality of the bivalves Nodipecten nodosus and wa- 214
ter in mariculture farms in the coves of Ilha Grande Bay, Angra dos Reis, RJ (Figure 1), 215
focusing on the prevalence of Pseudomonas spp. and Vibrio spp. over a year" this information was already presented.
Please refrain from using 3D graphs where possible, as they are far worse at showing the differences between studied groups than standard 2D ones.
4. Discussion
Different font sizes. At the same time, the topic of the article is "Studies of the Microbiological Quality of Nodipecten nodosus (Linnaeus, 1758) in Water from Farms for Human Consumption in Angra dos Reis, Rio de Janeiro, Brazil" and in discussion section we mostly see information centered about the presence of bacteria and chemical compounds in the waters. Without even presenting enough information on the main topic (or it would seem so) of the article which is once again - Studies of the Microbiological Quality of Nodipecten nodosus. In my opinion this demands either chaning the title of the article, or severe re-building the whole discussion section.
5.Conclusions
Once again information that has been presented numerous times, is being repeated here.
Overall I also suggest a round of English editing, as I cannot omit, the issues with quality of the language.
As such I can only recommend major revision.
Comments on the Quality of English Language
English should be improved in large sections of the manuscript, not only in terms of a number of mistakes made, and lack of clarity in some cases, but also due to the lack of seamless connection between some of the presented paragraphs.
Author Response
As a reviewer, I must point out that the manuscript's current state does not allow for publication in a scientific journal. I noticed several issues that need to be addressed first. At this point, it is very clear that the proposed manuscript lacks the proper polish and quality. Here, I will provide some observations for the specific chapters.
- Introduction: General observations: Some sections of the Introduction repeat the same information unnecessarily. See lines 63-64 and 55-56 as an example. In some cases, the text does not have perfect connections between paragraphs. The section on the organism's biology should be expanded to better highlight the reasons why its biology makes it a potential reservoir for potential pathogens and, next to them, ARGs.
R: Thank you, the requested corrections have been inserted.
- Methods section: General observation - Different font sizes. Unnecessary bolding of certain subchapters.
R: Thank you, the requested corrections have been inserted.
- Some descriptions need to be expanded as they are confusing - see line 143 as an example
Chapter names are being repeated. Line 122-á; Line 126 - needs to be corrected; Line 129 - in the area ; Lines 165 and 167 - there is no need to repeat the legislation twice.
R: Thank you
Nothing particularly concerning regarding the MALDI-TOF part of the studies.
R: The MALDI-TOF analysis enables the rapid and reliable identification of bacteria. Although other analyses can complement the data, MALDI-TOF provides sufficient information for an accurate assessment.
- Results: Once again, a section of the text was repeated, information - "This study evaluated the microbial quality of Nodipecten nodosus bivalves and water in mariculture farms in the coves of Ilha Grande Bay, Angra dos Reis, RJ (Figure 1), 215 focusing on the prevalence of Pseudomonas spp. and Vibrio spp. over a period of one year" this information has already been presented. Please avoid using 3D graphs whenever possible, as they are much worse at showing the differences between the studied groups than the standard 2D ones.
R; Thank you, the requested corrections have been inserted.
- Discussion: Different font sizes. The topic of the article, 'Studies on the Microbiological Quality of Nodipecten nodosus (Linnaeus, 1758) in Waters from Farms for Human Consumption in Angra dos Reis, Rio de Janeiro, Brazil,' is of great importance. In the discussion section, we see mainly information focused on the presence of bacteria and chemical compounds in the waters, without presenting enough info on the article's main topic (or it seems), which is, once again, Studies on the Microbiological Quality of Nodipecten nodosus. This requires changing the article's title or severely rebuilding the entire discussion section.
R: The title was changed to "Occurrence of Vibrio spp. and Pseudomonas spp. isolates of Nodipecten nodosus (Linnaeus, 1758) and water from a mariculture farm in Angra dos Reis, Brazil."
- Conclusions: Once again, information that has been presented numerous times is being repeated here. Overall, I also suggest a round of English editing, as I cannot omit the issues with the quality of the language. As such, I can only recommend a major revision.
R: Thank you for your efforts. The manuscript has been re-evaluated, and repetitions have been minimized and avoided.

Reviewer 3 Report
Comments and Suggestions for Authors
Review for the paper “Studies of the microbiological quality of Nodipecten nodosus (Linnaeus, 1758) in water from farms for human consumption in Angra dos Reis, Rio de Janeiro, Brazil” by Antonia Lucia Santos , Salvatore Giovanni De-Simone, Guilherme Silva Lourenço Carvalho , Kayo Cesar Bianco Fernandes , Maysa Beatriz Mandetta Clementino to “Microorganisms”.
The Lion's paw scallop is of particular economic significance in southeastern Brazil due to extensive farming activities in the region. A comprehensive analysis of its microbial diversity in aquaculture waters was conducted by the authors of the study, with a focus on potential microbiological contaminants that could compromise the safety of these scallops. The researchers evaluated the diversity of Lion's paw scallop gill arch tissues and the surrounding water of a scallop farm in Angra dos Reis, Rio de Janeiro, Brazil. Their findings revealed that the isolates exhibited varying degrees of susceptibility to antimicrobials, with experiments on Pseudomonas putida showing the highest percentage of resistance to ciprofloxin. All these isolates were sensitive to meropenem, while Vibrio alginolyticos and Vibrio fluvialis exhibited greater resistance to this antimicrobial.
The study's limitations include a lack of statistical analysis and the exclusion of spatial and temporal patterns. The discussion is inadequate, as it fails to offer significant contributions to the field.
Major flaws.
Abstract. The novelty of this study is not clear from the abstract.
Introduction. L 40-42. The authors should clarify the direct economic impact of Nodipecten nodosus and provide examples of its economic value, including more precise numbers or data on the economic value of Nodipecten nodosus globally and in Brazil (e.g., production volume or revenue).
Introduction. L42. More information about the ecological roles of this mollusk should be included in the text.
Introduction. L 55. The authors should briefly describe the nutritional benefits of scallops.
Introduction. L 55-62 and L 63-67. These two paragraphs are the same in terms of their content. The latter is required to be removed.
Introduction. L 74. It would be beneficial to provide data on the prevalence of foodborne illnesses caused by scallops, particularly those associated with pathogens such as E. coli and Salmonella, both globally and in Brazil.
Materials and Methods. L 123-157, 185-198. The authors should review this text, as it contains repetitions, such as the description of sampling locations, collection methods, and devise. Duplicated information should be removed.
Materials and Methods. L 124-125. The authors should clarify the criteria used to select the specific collection points (Jacuecanga, Gipoia Island, and Bananal Island).
Materials and Methods. L 166. The authors should explain why this specific size range (6.25 ± 1 cm) was chosen. Does this size have biological or economic significance?
Materials and Methods. L 179. The term "direct procedure" could benefit from a brief description of the steps involved in the direct procedure to enhance understanding.
Materials and Methods. The authors considered the scallop gills in the study. However, this tissue is typically discarded and not used for consumption. The authors' decision to omit the adductor muscle from the analysis warrants elucidation.
Results. L 218-219. The authors mention prevalence data in percentages, while Figure 2 shows absolute values. It is essential for the authors to maintain consistency in their text and figures.
Results. L 252. Please check the text.
Results. Sections 3.2-3.4. A statistical comparison of these data would provide valuable support for the main conclusions.
Results. Figure 5. The authors should provide the full data for the species of isolates.
Results. The authors did not study spatial and temporal patterns in the data.
Discussion. The discussion contains extensive text that does not have direct links to the results of the authors' study, while some authors' data was not discussed. For instance, there is an absence of comparisons regarding the prevalence and antimicrobial resistance among sites and across different seasons. The factors that contribute to the high resistance rates observed against certain drugs (e.g., ticarcillin-clavulanic acid, meropenem) were not discussed. How do environmental factors, such as temperature, salinity, and pollution, influence the prevalence and virulence of the common species?
Additionally, the practical implications of the authors' findings are not clearly highlighted.
This section should be rewritten to focus on the main findings and their importance for monitoring and public health in the regions.
Specific remarks
L 129. Consider replacing “na área” with “an area”
The English should be checked as the text contains some minor errors.
Author Response
The study's limitations include the lack of statistical analysis and the exclusion of spatial and temporal patterns. The discussion is inadequate, as it does not contribute significantly to the field.
Major flaws.
1) Abstract. The novelty of this study is not clear from the abstract.
R: Thank you, modifications have been made. "Ilha Grande, in Angra dos Reis, is the largest island in Rio de Janeiro and the sixth largest marine island. Its economy is based on fishing and mariculture, with emphasis on the mollusk Nodipecten nodosus. However, microbiological contamination can compromise its consumption. The rise of antimicrobial resistance in aquatic ecosystems is a global concern as these environments harbor resistant bacteria. This study evaluated the diversity and occurrence of Vibrio alginolyticus and Pseudomonas putida, opportunistic pathogens associated with infections in marine animals, in gill tissues and in the water of a farm in Angra dos Reis".
2) Introduction. L 40-42. Authors should clarify the direct economic impact of Nodipecten nodosus and provide examples of its economic value, including figures or more precise data on the economic value of Nodipecten nodosus globally and in Brazil (e.g., production volume or revenue).
R: A sentence weas introduced.
R: The scallop Nodipecten nodosus [1] (Pectinidae) is a bivalve of global importance in mariculture, including in Brazil [2-4] [20]. In 2022, production reached 1.74 million tons, mainly in China, the USA, and the EU, generating emissions of 4.3 billion dollars (2.8% of aquatic products).
Global warming and water pollution may have caused a drop in mollusk production in Ilha Grande Bay (RJ) [44], impacting the economy of Rio de Janeiro. According to a study by researchers from UFRJ, Uenf, and IED_BIG, published in September [43], the production of scallops (Nodipecten nodosus) for mariculture dropped from 51.2 tons in 2016 to 10.2 tons in 2022. Scallops are valued in haute cuisine worldwide.
3) Introduction. L42. More information about the ecological roles of this mollusk should be included in the text.
R: The mollusk microbiome plays a fundamental role in nutrition, digestion, metabolism, immunity, reproduction, and environmental adaptation, in addition to influencing their behavior and ecological relationships. This balance promotes a harmonious interaction between parasite and host [103], [104].
4) Introduction. L 55. The authors should briefly describe the nutritional benefits of scallops.
R`: A paragraphy was introduced
5) Introduction. L 55-62 and L 63-67. These two paragraphs are the same in terms of content. The last one should be removed.
R: Thank you, the requested corrections have been inserted.
6) Introduction. L 74. It would be beneficial to provide data on the prevalence of foodborne illnesses caused by scallops, particularly those associated with pathogens such as E. coli and Salmonella, both globally and in Brazil.
R: Thank you, the requested corrections have been inserted.
7) Materials and methods. L 123-157, 185-198. The authors should review this text, as it contains repetitions, such as describing sampling sites, collection methods, and devices. Duplicate information should be removed.
R: Thank you, the requested corrections have been inserted.
8) Materials and methods. L 124-125. The authors should clarify the criteria for selecting the specific collection points (Jacuecanga, Ilha da Gipoia, and Ilha do Bananal).
R: Thank you, this was done. "The criteria for selecting points for the three specific collections was based on the different impacts observed on scallop mortality: Gipóia Island, where there was no mortality; Banana Cove, site of mass mortality; and scallop cultivation, where high larval mortality was recorded."
9) Materials and methods. L 166. The authors should explain why this specific size range (6.25 ± 1 cm) was chosen. Does this size have biological or economic significance?
R: Thank you. The requested corrections have been made. This size represents the mean of the sample group. Furthermore, the scallops are supplied by shellfish farmers who donated only the smallest ones for the study due to their high economic value.
10) Materials and methods. L 179. The term "direct procedure" could benefit from a brief description of the steps involved in the direct procedure to improve understanding.
R: Thank you, this was done. "The isolated strains were cultivated in BHI Agar at 350C for 24 h to prepare the samples. Then, a colony of each isolate was chosen, and, using the direct transfer procedure (fishing each with a loop), the samples were applied to each spot on the microplate (96 MSP, Bruker® —Billerica, USA)".
11) Materials and methods. The authors considered scallop gills in the study. However, this tissue is normally discarded and not used for consumption. The authors' decision to omit the adductor muscle from the analysis justifies the elucidation.
R: Filtration of macerated scallop tissues and collected seawater was carried out by a vacuum system using

Round 2
Reviewer 2 Report
Comments and Suggestions for Authors
Dear Authors,
The article at this point, still does not fully meet the standards necessary for publication. First issue, starts just at the beginning - Abstract. It should present condensed "substance" of the article, with inclusion of some results, conclusion etc. Secondly in the body of the manuscript (also in the new parts of the text) there are some typos present.
Comments on the Quality of English LanguageAs mentioned in the previous section, the article still needs some corrections in this area.
Author Response
The article at this point, still does not fully meet the standards necessary for publication. First issue, starts just at the beginning –
- It should present condensed "substance" of the article, with the inclusion of some results, conclusion etc.
R: Thank you for your constructive comments on our manuscript. The abstract, introduction and the discussion were modified according to the instructions of the two reviewers
- Secondly in the body of the manuscript (also in the new parts of the text) there are some typos presents.
R: Thank you, all the text was rechecked.

Reviewer 3 Report
Comments and Suggestions for Authors
Second review for the paper “Studies of the microbiological quality of Nodipecten nodosus (Linnaeus, 1758) in water from farms for human consumption in Angra dos Reis, Rio de Janeiro, Brazil” by Antonia Lucia Santos, Salvatore Giovanni De-Simone, Guilherme Silva Lourenço Carvalho, Kayo Cesar Bianco Fernandes, Maysa Beatriz Mandetta Clementino to “Microorganisms”.
Although the authors have addressed some of my comments, most recommendations were not considered and concerns remain unexplained including the following:
Materials and Methods. The authors considered the scallop gills in the study. However, this tissue is typically discarded and not used for consumption. The authors' decision to omit the adductor muscle from the analysis warrants elucidation.
Results. L 242-245. The authors mention prevalence data in percentages, while Figure 2 shows absolute values. It is essential for the authors to maintain consistency in their text and figures.
Results. Sections 3.2-3.4. A statistical comparison of these data would provide valuable support for the main conclusions.
Results. Figure 5. The authors should provide the full data for the species of isolates.
Results. The authors did not study spatial and temporal patterns in the data.
Discussion. The discussion contains extensive text that does not have direct links to the results of the authors' study, while some authors' data was not discussed. For instance, there is an absence of comparisons regarding the prevalence and antimicrobial resistance among sites and across different seasons. The factors that contribute to the high resistance rates observed against certain drugs (e.g., ticarcillin-clavulanic acid, meropenem) were not discussed. How do environmental factors, such as temperature, salinity, and pollution, influence the prevalence and virulence of the common species?
Additionally, the practical implications of the authors' findings are not clearly highlighted.
This section should be rewritten to focus on the main findings and their importance for monitoring and public health in the regions.
Author Response
Second review for the paper "Studies of the microbiological quality of Nodipecten nodosus (Linnaeus, 1758) in water from farms for human consumption in Angra dos Reis, Rio de Janeiro, Brazil" by Antonia Lucia Santos, Salvatore Giovanni De-Simone, Guilherme Silva Lourenço Carvalho, Kayo Cesar Bianco Fernandes, Maysa Beatriz Mandetta Clementino to "Microorganisms".
Although the authors have addressed some of my comments, most recommendations were not considered and concerns remain unexplained including the following:
1)Materials and Methods. The authors considered the scallop gills in the study. However, this tissue is typically discarded and not used for consumption. The authors' decision to omit the adductor muscle from the analysis warrants elucidation.
R: Thank you for your constructive comments on our manuscript.
2) Results. L 242-245. The authors mention prevalence data in percentages, while Figure 2 shows absolute values. It is essential for the authors to maintain consistency in their text and figures.
R: Thank you. These observations were accepted, and the text was correct,
3) Results. Sections 3.2-3.4. A statistical comparison of these data would provide valuable support for the main conclusions.
R: There is no doubt that the application of statistics could improve the work, but I believe that it would not change the qualitative results. At this time, we have no way of recovering the data, as it is stored at BGI, which is more than 300 km away from where we are, and the journey requires time and resources.
4) Results. Figure 5. The authors should provide full data for the species of isolates.
R: Thank you. These omissions have been corrected.
5) Results. The authors did not study spatial and temporal patterns in the data. The authors did not study spatial and temporal patterns in the data.
R: These points are important and would contribute more to the work. However, they were not foreseen when the study was carried out. Despite this, they did not interfere with the results, the objective of which was to confirm the contamination and the resistance of the pathogens.
6: Discussion. The discussion contains extensive text that does not have direct links to the results of the authors' study, while some authors' data was not discussed. For instance, there is an absence of comparisons regarding the prevalence and antimicrobial resistance among sites and across different seasons. The factors contributing to the high resistance rates observed against certain drugs (e.g., ticarcillin-clavulanic acid, meropenem) were not discussed. How do environmental factors, such as temperature, salinity, and pollution, influence the prevalence and virulence of the common species?
R: The introduction and discussion were completely redone according to the suggestions of the two reviews. Their length was shortened, and repeated sentences were removed.
7) Additionally, the practical implications of the authors' findings are not highlighted. This section should be rewritten to focus on the main findings and their importance for monitoring and public health in the regions.
R: The corrections made aimed to include objectivity and focus.
